# The Interaction between B87 Vaccine Strain and BC6/85 of Infectious Bursa Disease Virus in SPF Chickens

**DOI:** 10.3390/v14102111

**Published:** 2022-09-23

**Authors:** Ling Chen, Xiaoyue Yang, Yafen Song, Taozhen Jiang

**Affiliations:** 1Department of Veterinary Culture Collection and Detection, China Institute of Veterinary Drug Control, No. 8 South Street, Zhongguancun, Haidian District, Beijing 100081, China; 2College of Veterinary Medicine, South China Agricultural University, No. 483 Wushan Road, Wushan Street, Tianhe District, Guangzhou 510642, China

**Keywords:** IBDV, B87 and BC6/85, virus quantification, viral interference

## Abstract

This study was initiated to determine the interaction between two infectious bursal disease virus (IBDV) strains in the early stages of infection by detection and quantification of IBDV RNA in lymphoid and non-lymphoid tissues. SPF chickens were inoculated with single infection or dual infection by the mild strain B87 followed by the pathogenic strain BC6/85 at 0, 1, 2, and 3 days post-inoculation (dpi) with B87. Real-time RT-PCR assays were developed to examine the viral loads of the tissues collected at various time intervals. The results reveal that B87 could delay the time point of positive detection of the BC6/85 strain in the bursa of Fabricius from 1 dpi to 3 dpi, indicating that B87 interfered with the replication of BC6/85. The interference occurred when BC6/85 was inoculated at 2 dpi and 3 dpi with the B87 strain. Moreover, BC6/85 could affect the proliferation and duration of B87 in SPF chickens. The rates of positive detection for B87 decreased significantly during dual infection. The investigation of the interaction between the two strains is important for the implementation of appropriate control measures.

## 1. Introduction

Infectious bursal disease (IBD) is an acute, highly contagious immunosuppressive disease of young chickens with worldwide prevalence. The disease is caused by the IBD virus (IBDV) in the genus *Avibirnavirus* of the family *Birnaviridae*. Chickens are the only animals known to develop the clinical disease with distinct lesions. The highest susceptibility to acute IBD occurs in chickens between 3 and 6 weeks of age. IBD is of great economic importance owing to its high mortality rate and prolonged immunosuppression in young chickens [1]. Classical IBDV induces approximately 10–50% mortality, whereas the very virulent IBDV induces approximately 50–100% mortality with typical signs and lesions [2].

In China, a strict vaccine immunization program is applied to prevent and control IBD [3]. B87 is the most widely used live-attenuated vaccine strain. The widespread use of vaccines exerts high selection pressure, and outbreaks of IBD occur sporadically in some immunized farms [3,4,5,6]. Poultry farms are likely to be exposed to vaccine and field strains [7]. Therefore, it is necessary to investigate the interactions between the vaccine and pathogenic strains of the virus.

Many studies have addressed the persistence and distribution of IBDV infection in tissues [8,9,10,11]. The present study extends these findings by co-infection of a mild and a pathogenic IBDV and the quantitative evaluation of the degree of such interference. Real-time RT-PCR assays were developed to examine the viral loads of the tissues collected at various time intervals. The high sensitivity and specificity of real-time RT-PCR make it an ideal assay for establishing in vitro and in vivo model systems in which different viruses and their interactions can be studied [12].

This study was conducted to obtain information on viral interference between B87 and BC6/85 during mixed and sequential infections as compared with the results of single viral infection in vivo. The objectives of this study were to investigate the possible causes of IBD immune failure and provide guidance for clinical application.

## 2. Materials and Methods

### 2.1. Chickens

Twenty-one-day-old SPF chickens were purchased from Beijing Boehringer Ingelheim Vital Biotechnology Co., Ltd., Beijing, China.

### 2.2. Virus Strains

B87 strain (CVCC AV140), the master seed of a live virus vaccine with intermediate virulence, was diluted to a titer of 10^3.0^ ELD_50_/0.1 mL and administered orally. BC6/85 strain (CVCC AV7), the standard challenge strain, was diluted to 10^3.0^ ELD_50_/0.1 mL and was administered via the oral route. Both strains were supplied by the China Veterinary Culture Collection Center (CVCC).

### 2.3. Primers and Probes

Sequence alignments of B87 and BC6/85 were conducted using the Clustal algorithm implemented in MEGA 5.0 (Sinauer Associates, Inc., Sunderland, MA, USA). Primers and TaqMan probes were designed and synthesized by Shanghai Invitrogen Biotechnology Co., Ltd, Shanghai, China (Table 1).

### 2.4. Specificity Testing

A BLAST search was performed to evaluate the occurrence of non-specific homology with other IBDV genome regions and the chicken genome. Previously confirmed IBDV-negative samples were also processed. Cross-reaction between specific probes and other common RNA viruses (Infectious Bronchitis Virus, Newcastle Disease Virus, and Avain Influenza Virus) was evaluated.

### 2.5. The Generation of Standard Curve and Estimation of ELD_50_ Values

To generate standard curves for B87 (P1) and BC6/85 probe (P2), serial 10-fold dilutions of the IBDV B87 and BC6/85 strains were used, and total RNA extracts were analyzed in three independent runs by real-time RT-PCR. Standard curves for each probe were generated by plotting threshold cycle (Ct) values per three replicates per standard dilution versus the logarithms of ELD50 values to determine analytical sensitivity of the assays. The amplification efficiency was calculated using the equation E = (10^(−1/k)^) − 1, where (k) is the slope of the linear regression line [13,14]. A value of 1 corresponds to 100% amplification efficiency. The coefficients of variation (CVs) of Ct values were assessed separately for each standard dilution by analyzing three replicates of the same analytical run [15].

### 2.6. Experimental Design

Experiments were conducted with 21-day-old SPF chickens. A total of 124 birds were randomly divided into 7 groups (16 birds in each group of 1–6, 28 birds in group 7). Samples were collected from lymphoid tissues (bursa and spleen) and non-lymphoid tissues (blood, thigh muscle, kidney, and liver). Two chickens per group were euthanized at each designated time point. All samples were stored at −80 °C until used for RNA extraction. The schedules of inoculation and sample collection are shown in Table 2.

### 2.7. Viral Load Detection by Real-Time RT-PCR

#### 2.7.1. Tissue Processing

Tissue samples (1 g each) including bursa, spleen, thigh muscle, kidney, and liver were collected using separate sterile surgical scissors and forceps. Each sample was suspended in 1 mL sterile phosphate-buffered saline in 2.0 mL sterile tubes. The tissues were homogenized using a freezing grinder (Hoder, Miyun, Beijing, China) at 65 Hz for 2 min at 4 °C. The homogenized tissues were processed through three cycles of freeze–thaw at −70 °C and then subjected to centrifugation at 5000 rpm for 20 min at 4 °C. A total of 200 µL of supernatant of each homogenized sample was used for RNA extraction. Blood samples were directly used for RNA extraction without further process.

#### 2.7.2. RNA Extraction and Real-Time RT-PCR

RNA was extracted with a Takara MiniBest Viral RNA/DNA Extraction Kit Ver. 5.0 (Takara Bio., Changping, Beijing, China) according to the manufacturer’s instructions. The extracted RNA was resuspended in 30 µL nuclease-free water.

For the RT step, the SuperScript III First-Strand Synthesis System for RT-PCR kit (ThermoFisher Inc., Waltham, MA, USA) was used in a 20 µL final reaction volume.

Real-time PCR was conducted in a 20 µL reaction containing 1 x TaqMan Master Mix (Applied Biosystems), 1 µM of each primer, 0.2 µM probe, and 5 µL cDNA. Thermocycling was performed on an ABI 7500 (Applied Biosystems) and consisted of 5 min incubation at 50 °C and denaturation for 10 min at 95 °C, followed by 40 cycles of 15 s at 95 °C and 1 min at 60 °C and a final step of 5 min at 70 °C. Fluorescence measurements from specific reporter fluoropores were collected at the 60 °C step of each cycle.

### 2.8. Statistical Analysis

All presented results are shown as the mean ± standard deviation (SD)/coefficient of variation (CV). Statistical analysis was conducted using GraphPad Prism 7.04 software (San Diego, CA, USA). Statistical analysis of the viral loads in group 1 and group 2 were performed using the Mann–Whitney test. *T*-tests were used to evaluate whether the difference of the viral loads between group 1 and group 2 was statistically significant. A *p*-value of less than 0.05 was considered to be statistically significant.

### 2.9. Ethics Statement

The animal experiments in this study were performed in biosafety level 2 facilities at the China Institute of Veterinary Drug Control (IVDC) in accordance with animal welfare guidelines and applicable laws and were approved by the Ethics Committee of IVDC (201600182; 14 September 2016)

## 3. Results

### 3.1. Development and Validation of Real-Time RT-PCR Assays

To discriminate and quantify B87 and BC6/85 separately, two TaqMan probes were designed and named P1, which was labeled with the fluorescent dye FAM and was specific for the B87 strain, and P2, which was labeled with the fluorescent dye VIC and was specific for the BC6/85 strain.

The performance of the P1 and P2 probes was determined via separate assays. Two standard curves were generated using 10-fold serial dilutions of ELD_50_ value standards. The P1 probe showed a linear dynamic range of 10^0.8^–10^4.8^ ELD_50_ value/reaction, with an average R^2^ of 0.9924 and an efficiency of 92%. The P2 probe showed a linear dynamic range between 10^0.2^ and 10^5.2^ ELD_50_ value/reaction, with an average R^2^ of 0.9926 and an efficiency of 98% (Figure 1). Assay reproducibility was assessed by obtaining inter-assay CVs for each standard ELD_50_ value dilution of B87 and BC6/85 (Table 3); the ranges of inter-assay CV% for standard ELD_50_ value dilution of B87 and BC6/85 were 0.65–1.69 and 0.98–3.43, respectively.

All tissue samples in group 7 (negative control group) were tested with P1 and P2 separately. The Ct values for P1 were higher than 33, and the Ct values for P2 were higher than 32.

### 3.2. Viral Loads of Single and Mixed Viral Infection Groups

In Group 1, B87 was inoculated singly. Bursa samples were positive 3–7 days post-inoculation (dpi). Blood samples were negative 1 dpi and 5 dpi and positive at other times. Kidney samples were positive 2–6 dpi. The results for liver and spleen tissues are irregular. The muscle samples were positive at all time points except for 5 dpi. Except for the bursa of Fabricius, viral loads of other tissues were all low. The results show that the viral load of B87 in the bursa of Fabricius was higher than in other tissue samples (Figure 2A).

In Group 2, BC6/85 was inoculated singly. Bursa samples 1–7 dpi were positive. Blood samples 1 and 4 dpi were negative and were positive at other time points. Kidney samples were positive 3–7 dpi. Liver and muscle samples were all positive except 1 dpi. Spleen samples 2–7 dpi were positive (Figure 2B). Except for the bursa of Fabricius, viral loads of the other five tissues were higher than that of B87 (Table 4).

In Group 3, B87 and BC6/85 were dually inoculated. The detection results of BC6/85 in all tissues (except for blood) are consistent with those in Group 2, while the positive rates of B87 are significantly lower than those in Group 1. The results show that the replication of B87 in the tissues was interfered by BC6/85. The details are given in Figure 2C,D.

Different lowercase superscript letters within each column indicate significant differences between Groups 1 and 2.

### 3.3. Viral Loads of BC6/85 in Sequential Viral Infection Groups

In Group 4, SPF chickens were inoculated with B87 1 day prior to being inoculated with BC6/85. The positive rates of BC6/85 (Figure 3B) in the bursa of Fabricius, kidney, and liver were consistent with that in Group 2. There was little interference with the replication of BC6/85 in SPF chickens.

In Groups 5 and 6, the time points of positive detection of BC6/85 (Figure 3C,D) in the bursa of Fabricius were delayed from 1 to 3 dpi, and the results of liver and kidney are irregular. The results indicate that when B87 was inoculated 2 days or 3 days earlier than the infection of BC6/85, there was a negative effect on the virus replication of BC6/85 in chickens.

### 3.4. Viral Loads of BC6/85 in Bursa, Spleen, Kidney, and Liver

Whether the inoculation of the B87 strain was at the same time with BC6/85 (group 3) or 1 day earlier than the infection of BC6/85 (group 4), it had no significant effect for the proliferation and duration of the BC6/85 strain in the bursa of Fabricius (Figure 4A). However, when the inoculation of the BC6/85 strain was 2 days (group 5) or 3 days (group 6) later than the infection of the B87 strain, it could delay the time point of the positive detection of the BC6/85 strain in the bursa of Fabricius from 1 dpi to 3 dpi; the positive detection rates in liver (Figure 4C) and kidney (Figure 4D) decreased significantly, with the positive results being irregular. For the spleen (Figure 4B), little difference was observed among the five groups, except the viral loads of BC6/85 2–5 dpi in group 3 were lower than other groups.

## 4. Discussion

In this study, specific and sensitive real-time RT-PCR assays for the detection and differentiation of the B87 and BC6/85 strains were developed for the first time, paving the way for rapid and effective detection in subsequent experiments [10,15,16]. The standard challenge strain BC6/85 and the vaccine strain B87 could be detected in the blood, kidney, spleen, liver, bursa of Fabricius, and thigh muscle, indicating that IBDV was distributed throughout the body after oral infection [8,9,17], and the virulent strain BC6/85 was more frequently detected in various tissues than B87.

The interaction between the two strains was indicated by the difference in virus distribution during single and dual infection. The time interval between inoculations had a significant effect on the replication of the second inoculated virus. During dual infection, the B87 virus interfered with the BC6/85 virus which was inoculated 2 days later by inhibiting its replication in the bursa of Fabricius. The time point of positive detection results in the bursa of Fabricius for BC6/85 was delayed from 1 dpi to 3 dpi. The positive rates of virus detection in the kidney and liver were significantly reduced, and the positive results are irregular. The results show that strain B87 could not completely prevent the replication of BC6/85 in chicken tissues but could affect its dynamic distribution. In a previous study [18], the most significant interference occurred at 1 dpi. The difference in the time of interference could be due to the detection methods (real-time PCR and conventional PCR) or to the use of different strains.

The contribution of humoral immunity to defense against disease has been well-documented, as indicated by protection conferred by maternal antibodies [1]. Serum antibody responses of moderately virulent IBDV inoculated at 7 dpi were detectable by the agar gel precipitin test [19] and at 8 dpi by an antibody capture enzyme-linked immunosorbent assay [18]. When the time interval of infection between the two strains was reduced to 3 days, or even 1 day, antibody protection was unlikely to play a role in the early stages. Interference with the replication of BC6/85 by B87 might indicate competition for receptors on tissues [20] or an anti-viral immune response.

T-cell immunity may play an important role in defense against IBDV. Kim et al. [21] found that there was an increase in the numbers of intrabursal T-cells in IBDV-infected chickens, and the intrabursal T-cells were activated with increased levels of Ia and CD25 expression and increased production of IL-6, NOIF, and IFN-γ. Bursal T-cells proliferated in vitro upon stimulation with purified IBDV in a dose-dependent manner, whereas virus-specific T-cell expansion was not detected in the spleen. The results suggest that intrabusal T-cells and T-cell-mediated responses might be important for limiting virus replication in the bursa of Fabricius, but the spleen was not affected. In this study, when B87 was inoculated 2 days or 3 days earlier than the BC6/85 strain, the replication of BC6/85 in the bursa of Fabricius was suppressed, while the distribution of BC6/85 in the spleen was not affected, which was in accordance with that of Kim et al. [21].

The current study also indicates that the virulent strain BC6/85 may interfere with the replication and tissue distribution of the vaccine strain B87 in chickens. The viral distribution of the B87 strain during dual infection was significantly different from that under single infection; the positive rate of virus detection was significantly reduced, and there was no clear pattern for the time point of positive samples in each tissue. This could be due to the higher replication efficiency of the virulent strain in target tissues [17]. Tsukamoto et al. [17] compared the replication efficiency of three IBDV strains in lymphoid tissues and demonstrated that there was some difference in IBDV replication efficiencies in bursa, spleen, thymus, and bone marrow. Virulent strains showed higher antigen titers and frequency throughout the experiment.

The vaccine strain B87 could reduce the replication of BC6/85 during mixed infection in SPF chickens, indicating that immune protection can be achieved as early as 2 days after inoculation; vaccination with B87 live vaccine has the potential to respond to outbreaks of IBD. However, for chickens exposed to more virulent strains of IBDV, vaccination shows lower immune response due to less antigenic stimulation, and the effectiveness of live vaccines would reduce.

## Figures and Tables

**Figure 1 viruses-14-02111-f001:**
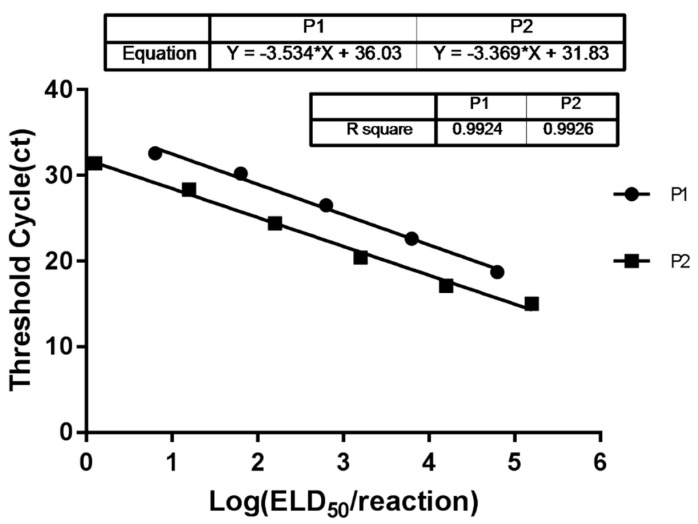
Standard curves of the developed real-time RT-PCR assay using B87 (P1) and BC6/85-specific probes (P2). Each point represents the mean Ct of three different measurements. The coefficients of determination R^2^ are indicated.

**Figure 2 viruses-14-02111-f002:**
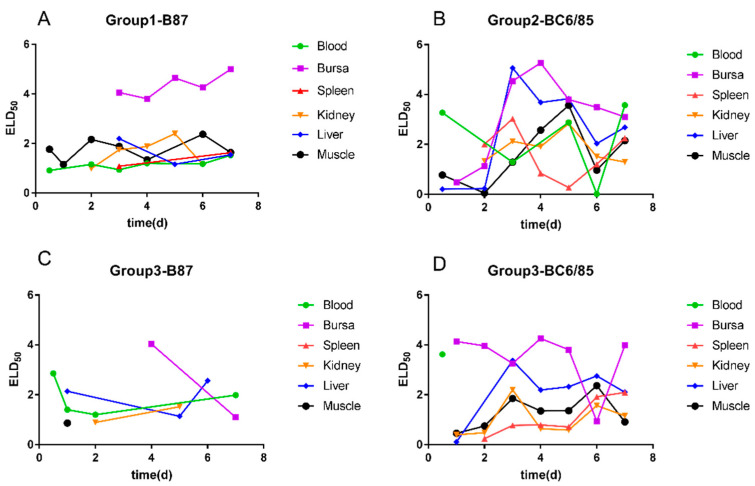
Viral loads of B87 and BC6/85 in bursa, spleen, liver, kidney, blood, and muscle of SPF chickens infected with B87 and (or) BC6/85. (**A**) B87 in Group 1; (**B**) BC6/85 in Group 2; (**C**) B87 in Group 3, spleen samples were all negative; (**D**) BC6/85 in Group 3.

**Figure 3 viruses-14-02111-f003:**
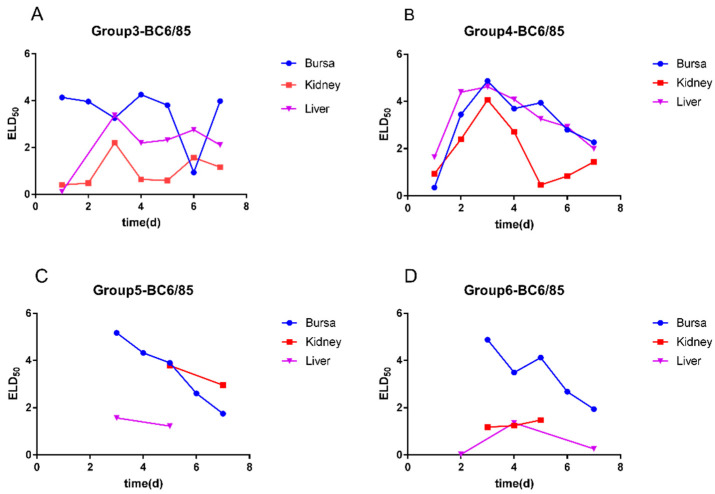
Viral loads of BC6/85 in the bursa of Fabricius, liver, and kidney of SPF chickens with sequential infection of B87 and BC6/85. (**A**) BC6/85 in Group 3, (**B**) BC6/85 in Group 4, (**C**) BC6/85 in Group 5, and (**D**) BC6/85 in Group 6.

**Figure 4 viruses-14-02111-f004:**
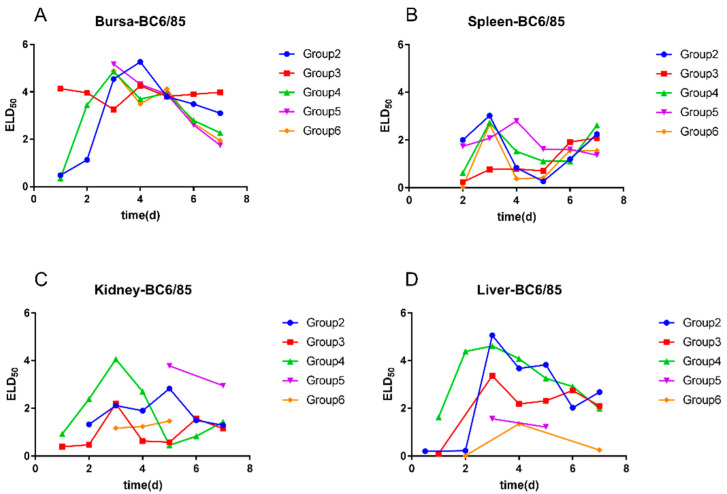
Viral loads of BC6/85 in the bursa of Fabricius, spleen, kidney, and liver of SPF chickens in different groups. (**A**) BC6/85 in the bursa of Fabricius, (**B**) BC6/85 in spleen, (**C**) BC6/85 in kidney, and (**D**) BC6/85 in liver.

**Table 1 viruses-14-02111-t001:** Primers and probes for B87 and BC6/85.

Primer/Probe	Sequence 5′→3′	Position ^a^	Amplicon Size
F1	CGACAACCTTATGCCATTCAATC	858–880	84 bp
R1	GTCACTATCTCCAGTTTGATGGATGT	916–941	
F2	GGGAGAGCTCGTGTTCCAAA	730–750	73 bp
R2	GATGGTGGCGCCCAGTAC	784–802	
P1	FAM-TGATTCCAACAAACGAGATAACCCAGCCA-TAMAR	884–912	
P2	VIC-AGCGTCCAAAGCAT-MGB	752–766	

^a^ Oligonucleotide position according to the strain B87 VP2 sequence.

**Table 2 viruses-14-02111-t002:** Schedules of inoculation and sample collection.

Group	0 H	12 H	24 H	36 H	48 H	60 H	72 H	84 H	96 H	120 H	144 H	168 H	192 H	216 H	240 H
1	B87 *	√	√		√		√		√	√	√	√			
2	BC6/85 *	√	√		√		√		√	√	√	√			
3	B87& BC6/85 *	√	√		√		√		√	√	√	√			
4	B87 *		BC6/85 *	√	√		√		√	√	√	√	√		
5	B87 *				BC6/85 *	√	√		√	√	√	√	√	√	
6	B87 *						BC6/85 *	√	√	√	√	√	√	√	√
7	/	√	√	√	√	√	√	√	√	√	√	√	√	√	√

* Each bird was orally inoculated with 0.1 mL containing 10^3.0^ ELD_50_ of inoculum; √ sampling.

**Table 3 viruses-14-02111-t003:** Variability of B87 and BC6/85 probes.

ELD_50_/Reaction	B87		ELD_50_/Reaction	BC6/85	
	Mean Ct	CV		Mean Ct	CV
10^0.1^	- ^a^		10^0.2^	31.40	1.98
10^0.8^	32.59	0.82	10^1.2^	28.35	0.98
10^1.8^	30.23	0.65	10^2.2^	24.43	2.63
10^2.8^	26.52	1.17	10^3.2^	20.41	3.43
10^3.8^	22.62	1.69	10^4.2^	17.13	3.03
10^4.8^	18.73	1.38	10^5.2^	15.01	2.60

CV, coefficient of variation of Ct values (%). ^a^ Ct value out of dynamic range.

**Table 4 viruses-14-02111-t004:** Mean viral loads (log ELD_50_) of Groups 1 and 2.

Organ	IBDV Strain	Day(s) Post-Inoculation							
		0.5	1	2	3	4	5	6	7
Bursa	B87	-	-	-	4.06 ± 0.22 ^A^	3.81 ± 0.32	4.64 ± 0.35	4.26 ± 0.25	4.99 ± 0.36
	BC6/85	-	0.48	1.13 ± 0.02	4.54 ± 0.47	5.27 ± 0.37	3.80 ± 0.22	3.49 ± 0.32	3.10 ± 0.28
Blood	B87	0.91 ± 0.03 ^a^	-	1.16 ± 0.05	0.94 ± 0.05 ^a^	1.20 ± 0.09 ^a^	-	1.18 ± 0.11 ^a^	1.53 ± 0.13 ^a^
	BC6/85	3.27 ± 0.22 ^b^	-	-	1.29 ± 0.07 ^b^	2.88 ± 0.22 ^b^	-	0.01 ^b^	3.57 ± 0.26 ^b^
Spleen	B87	-	-	- ^a^	1.08 ± 0.13 ^a^	-	-	- ^a^	1.63 ± 0.11
	BC6/85	-	-	2.01 ± 0.12 ^b^	3.03 ± 0.22 ^b^	0.84	0.27 ± 0.12	1.20 ± 0.12 ^b^	2.25 ± 0.17
Kidney	B87	-	-	0.99 ± 0.04	1.75 ± 0.18	1.88 ± 0.16	2.40 ± 0.18	1.15 ± 0.10	- ^a^
	BC6/85	-	-	-	2.12 ± 0.12	1.91 ± 0.18	2.84 ± 0.27	1.51 ± 0.20	1.29 ± 0.09 ^b^
Liver	B87	-	-	-	2.19 ± 0.20 ^a^	-	1.16 ± 0.17 ^a^	- ^a^	1.54 ± 0.11 ^a^
	BC6/85	0.21	-	0.23	5.07 ± 0.40 ^b^	3.68 ± 0.32	3.83 ± 0.32 ^b^	2.03 ± 0.18 ^b^	2.66 ± 0.22 ^b^
Thigh muscle	B87	1.77 ± 0.17	1.15 ± 0.12 ^a^	2.16 ± 0.21	1.88 ± 0.19	1.34 ± 0.16	- ^a^	2.37 ± 0.16	1.64 ± 0.11
	BC6/85	0.77	- ^b^	0.05	1.29 ± 0.12	2.57 ± 0.22	3.57 ± 0.29 ^b^	0.97	2.15 ± 0.20

-, Negative; ^A^ values are means of log ELD_50_ ± standard deviation; Different lowercase superscript letters within each column indicate significant differences between groups.

## Data Availability

Not applicable.

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
