# Peer review of "The Interaction between B87 Vaccine Strain and BC6/85 of Infectious Bursa Disease Virus in SPF Chickens"

_viruses, 2022, doi:10.3390/v14102111_

Round 1

Reviewer 1 Report

The authors investigated the interaction between the IBDV live-attenuated strain B87 and the IBDV pathogenic strain BC6/85 in SPF chickens through co-infection. The results revealed that B87 could delay the time point of positive detection of BC6/85 strain in the bursa of Fabricius from 24 to 72 hpi. Also, BC6/85 could affect the proliferation and duration of B87 in SPF chickens. The rates of positive detection for B87 decreased significantly during dual infection. Over all, the MS was well written, and the conclusion was supported by the data collected. The findings in the MS provide insights into the field for generating efficient immune preventive strategies for IBDV. Also, I have the following suggestion and question which should be addressed for the revised version.

1) In abstract:hours post-inoculation (PI)” suggested to be “hours post-inoculation (hpi), days PI suggested to be ”dpi (day post infection)

2) In introduction, genus Avibirnavirus of the family Birnaviridae should be genus Avibirnavirus of the family Birnaviridae . in vitro and in vivo should be in vitro and in vivo

3) In materials and methods, IBV, NDV, and AIV“ suggested to be given with full name.

4) sample collection are shown in Table 2 should be sample collection were shown in Table 2

5) All presented results are shown as should be All presented results were shown as

6) For the Statistical analysis, please indicate the method used for p-value assay.

7) the ct values should be the Ct values

8) Did the author perform the cross-reaction for B87 and BC6/85 using the P1 and P2 probes designated ?

9) Please briefly describe the data in Table3 for the Assay reproducibility

10) Table 4 might be not necessary.

11) The description for the data in Fig2,Fig3 suggested to be refined. The A,B,C,D should be labelled in Fig2 and Fig4. The negative data for spleen in Group3 should be also included in Fig2. G4,G5,G6 should be Group4, Group5,Group6 in Fig3.

12) In line 255 imune protection should be immune protection

13) In this study, could the inoculation of single BC6/85 cause the mortality of the infected chickens (Any infected chicken was died?)?. Could the prior infection of B87 reduce the mortality or the virus shedding in the cloaca caused by BC6/85 ?

Reviewer 2 Report

Manuscript ID: viruses-1928103

The Interaction between B87 Vaccine Strain and BC6/85 of Infectious Bursa Disease Virus in SPF Chickens

Comments to the Author:

Infectious bursal disease (IBD) is an acute, highly contagious immunosuppressive disease. In China, a strict vaccine immunization program is applied to prevent and control IBD. However, the high selection pressure by using of vaccine may cause the outbreaks of IBD sporadically in some immunized farms. The authors reveal an interesting phenomenon that the vaccine strain B87 interferes the replication of the virulent strain BC6/85 at different infection time point. There are some points listed below:

Minor comments:

1. Figure 2 and Figure 4, please mark the A, B, C and D in the picture corresponding to the figure captions.

2. Line173 and 174, log ELD50 should be written as log ELD50.

3. Figure 3, it was suggested that the abbreviation of G4, G5 and G6 should be given full name like the “Group3”, otherwise it should be noted in the figure captions.
